# Mapping the Typographic Latent Space of Digits

**Alayt Issak, Sarthak Kakkar, Nik Brown & Casper Harteveld**
Northeastern University
Boston, MA 02120, USA
{issak.a, kakkar.sa, nik.brown, c.harteveld}@northeastern.edu

**Sair Goetz**
Independent Typographer
Boston, MA
sairgoetz@gmail.com

## Abstract

Since the advancement of handwritten text to typefaces on a computer, the human mind has evolved towards corresponding various typefaces as norms of comprehension. Current-day typefaces, much like those written by hand, exist in disparities and are governed by consensus reached among Typographers. Currently, the PANOSE system, developed in 1998, is the most widely used and accepted method for classifying typefaces based on 10 visual attributes. In this work, we employ Disentangled Beta-VAE's, in an unsupervised learning approach, to map the latent feature space with a dataset of MNIST Style Typographic Images (TMNIST-Digit) of 0-9 digits across 2990 unique font styles. We expose the learning representation across a variety of font styles to enable typographers to contemplate and identify new attributes to their classification system.

## 1 Introduction

Typography has been evident since hieroglyphics and replicating text via moveable type has also existed via woodblock printing during the Tang Dynasty (618–907 AD) in China. In the Latin Type, typefaces emerged as calligraphy was overtaken by the incentive for mass production, and in the history of Western printing, was inaugurated by the letterpress in 1450 (Forrest, 2021).

In its ontology, Latin typefaces are mostly named after the people that devised them, such as Garamond in France, or the movements that inspired them, such as Roman (the foreground to Times New Roman) in Italy (Wells, 2022). In later formalization, a serif (a projection on the stroke of a letter) was then introduced to mitigate the confusions one might have between differentiating nuances, such as the number '1' from the letter 'l', and existing mechanisms that classify typefaces into five categories present as follows — Serif, Sans-Serif (without serif), Script, Decorative and Blackletter (Priya, 2019).

However, we find that the exact consensus to the patterns observed from the series of heuristics and matriculation is unknown and lacks comprehension. This is especially important as PANOSE, the current industry standard for classifying typefaces based on 10 attributes (Stevahn, 1996), inherits the aforementioned historical design process with explicit discrepancies. For example, in one study, Helvetica, a sans-serif font type, was found to be successful for readers with dyslexia, whereby fonts with serifs were found to decrease participants' reading performance (Rello & Baeza-Yates, 2013), yet in another study, the results were the opposite and quite encouraging of serifs (Hillier, 2008).

This questions the nature of discovery for typefaces to begin with, as the addition of serifs provides disparate affordances; that is, an actual dyslexic typeface from the ground up is not addressed to mitigate readability. Notable typographers, such as Sofie Beier, have equally signaled that legibility varies significantly depending on the reading situation (Beier et al., 2018), hence leading heuristics into question and the need for an overall alternate to the derivation for typeface mechanisms.

## 2 METHOD

In this paper, we seek to expose the typographic latent space of an encoder to represent typeface letter forms so typographers may identify attributes separate from those outlined by PANOSE. We utilize Disentangled Beta-Variational AutoEncoders ($\beta$-VAE) as our architecture to generate typefaces across a range of learned attributes due to its robust learning of disentangled representations (Burgess et al., 2018).

To specify our methodology, we choose $\beta$-VAE as our base architecture, which tunes $\beta$ on the Kullback–Leibler (KL) divergence term to emphasize the distance from the generated and ground truth probability distributions and learn statistically independent latent factors (Wei et al., 2020). We then particularly utilize a Disentangled $\beta$-VAE (Equation 1) as progressively tuning the control capacity ($C$) further exasperates the information capacity of the latent space to map the optimal value of the KL divergence term for the ELBO we aim to approach.

$$\mathcal{L}(\theta, \phi; x, z, \gamma, C) = \mathbb{E}_{q_\phi(z|x)}[\log p_\theta(x|z)] - \gamma |D_{KL}(q_\phi(z|x)||p_\theta(z)) - C| \tag{1}$$

Testing our hypothesis on digits, we utilize the TMNIST-Digits dataset (Magre & Brown, 2022) which consists of 29,900 examples, with 10 digits of MNIST-style images (0-9) for each of the 2900 font styles. We split our dataset into a $80, 10, 10$ train, validation, and test split, and train our model with the Pythae Disentangled Beta-VAE library unified by Chadebec et al. (2022) for 20 epochs with a learning rate of $10^{-4}$, capacity control ($C$) of 30 and a batch size of 64. We utilize Adam to update our weights and employ early stopping with a patience level of 5. Upon obtaining 16 as our optimal dimension for the latent space (similar to MNIST), we visualize samples from a Gaussian Mixed Model and Standard Normal distribution of the latent space learned by the Disentangled $\beta$-VAE below. We also present a reconstruction of a ground truth image to check our fidelity.

## 3 RESULTS

The latent space exhibits varying fidelity in our model's ability to encode representations across the 10 digits below. In Figures 1 (a), the digits 2 and 4 seem to learn well whereas $0, 8$, and 9 seem to be intertwined as per the poor reconstruction in the rightmost column of Figure 1 (c) and lose encoding in the latent space of both samples (Figures 1 (a) and (b)). In doing so, we present this work so typographers may investigate current heuristics to tailor their font types and attribute distinctions within these select digits, as per the deduction of letter forms extracted within these samples.

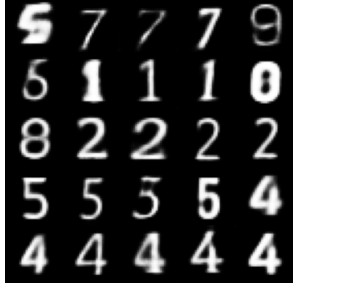 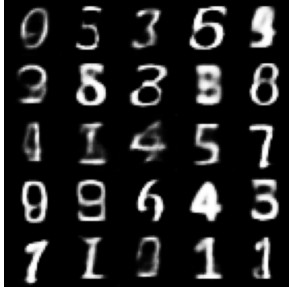 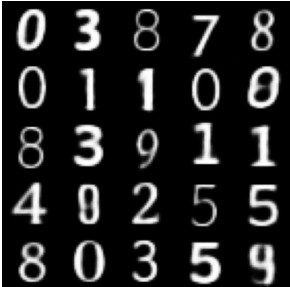

Figure 1: Sampling the Latent Space via (a) Gaussian Mixed Model and (b) Normal Distribution, along with (c) Reconstruction on the test set.

## 4 CONCLUSION

We find that mapping the "learning" that goes on in a model is a method to reverse the current heuristics by presenting the features the model has learned. Likewise, we present this endeavor to typographers so they may use these findings as a representation for unmasking current attributes of industry-standard font classification and matching systems. In future work, we seek to extend our findings toward letters, non-Latin typefaces, and the overarching question of typeface legibility.

ACKNOWLEDGEMENTS

We thank Nimish Magre for the insightful comments, guidance and curating the TMNIST dataset.

URM STATEMENT

The authors acknowledge that at least one key author of this work meets the URM criteria of ICLR 2023 Tiny Papers Track.

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
