# OpenReview forum: "Mapping the Typographic Latent Space of Digits"
_ICLR.cc/2023/TinyPapers — Submitted to Tiny Papers @ ICLR 2023_

### Official Review · Reviewer_N99x · 2023-03-21

**Confidence:** 4

**Summary Of Contributions:**

This paper describes an application of Variational AutoEncoders to explore the subtleties of different typefaces of numerical digits. The authors claim that understanding the latent space of VAEs can lead to advances in the classification of typefaces and fonts.

**Rating:**

High Potential (HP): a submission which meets the reviewing criteria and has potential to make an impact on the field

**Strengths And Weaknesses:**

Strengths:
1) The paper is well-written, well-motivated, and articulate.

Weaknesses:
1) Results section could be made more elaborate.

**Suggested Changes:**

1) A 2D visualization of the 16D latent space (after reduction using PCA or any dimensionality reduction technique) would show how the different typefaces are distributed.
2) The caption of Figure 1 could be made more descriptive so that it is more clear to the reader what the 3 images represent.

---

### Official Review · Reviewer_7Gmo · 2023-03-28

**Confidence:** 3

**Summary Of Contributions:**

Training a Disentangled Beta-VAE to map the latent feature space with the TMNIST-Digit dataset. The contributions of this paper may help typographers to consider and identify new letters/symbols in their classification system.

**Rating:**

Clear, Correct, and Reproducible (CCR): a submission which meets the reviewing criteria

**Strengths And Weaknesses:**

S: Application work of Beta-VAE to an important real-world problem. The paper is well-written and structured. Information about the method used, the results achieved, and a concise conclusion was provided.

W: No mention of open-sourcing the codes (e.g., anonymous GitHub repository).

**Suggested Changes:**

SC1. Please consider cutting some long phrases into a few small but concise phrases. For example, the second paragraph of the Methods Section is too large (a single phrase).

SC2. Please correct the sentence "In future work, we seek to further our limitations [...]". If there is space, please state one limitation of this work that you plan to investigate in future work.

SC3. As a public dataset has been used, I suggest open-sourcing the developed codes (e.g., GitHub) for improved reproducibility and transparency of this paper.

---

### Meta-Review · Area_Chair_ohHQ · 2023-04-05

**Recommendation:** Invite to present
**Confidence:** 3

**Metareview:**

This work employs a Disentangled Beta-VAE to map the latent feature space with a dataset of MNIST Style Typographic Images (TMNIST-Digit) of 0-9 digits across 2990 unique font styles. The proposed technique may help typographers to consider and identify new letters or symbols in their classification system.

In general, reviewers noted the strengths of application to a real-world problem, good writing, clear motivation, and concise conclusion. The paper can be further improved with more implementation details (or open sources), 2D visualization of the 16D latent space, and more careful proofreading.

**Summary:**

This work exposes the learning representation across a variety of font styles to enable typographers to contemplate and identify new attributes to their classification system.

**Comments And Feedback To The Authors:**

There are writing suggestions raised by the reviewers. Also, please handling the citations correctly when the references are as part of the text or not, i.e., using \citet{} for in-text references otherwise \citep{} with the ICLR template.

**Reason For Not Giving A Higher Recommendation:**

N/A

**Reason For Not Giving A Lower Recommendation:**

This paper is generally self-contained and meets the CCR reviewing criteria.

---

### Decision · Program_Chairs · 2023-04-08

Invite to present